# Unsupervised Conformal Inference: Bootstrapping and Alignment to Control LLM Uncertainty

## Abstract

Deploying black-box LLMs requires managing uncertainty in the absence of token-level probability or true labels. We propose introducing an unsupervised conformal inference framework for generation, which integrates: generative models, incorporating: (i) an LLM-compatible atypical score derived from response-embedding Gram matrix, (ii) UCP combined with a bootstrapping variant (BB-UCP) that aggregates residuals to refine quantile precision while maintaining distribution-free, finite-sample coverage, and (iii) conformal alignment, which calibrates a single strictness parameter $\tau$ so a user predicate (e.g., factuality lift) holds on unseen batches with probability $\geq 1 - \alpha$. Across different benchmark datasets, our gates achieve close-to-nominal coverage and provide tighter, more stable thresholds than split UCP, while consistently reducing the severity of hallucination, outperforming lightweight per-response detectors with similar computational demands. The result is a label-free, API-compatible gate for test-time filtering that turns geometric signals into calibrated, goal-aligned decisions.

## 1 Introduction

Reliable *uncertainty quantification (UQ)* for large language models (LLMs) is needed for trustworthy AI. An assertive yet baseless claim can swiftly spread and cause damage, but for most practitioners, frontier models arrive only as black-box APIs with no access to gradients, exact log probabilities, or hidden states (Lin et al., 2024). Hence deployment teams must make keep-or-discard decisions from samples alone.

In black-box deployments, LLM uncertainty must be inferred from the sampled outputs themselves. Query-only signals include: (i) semantic-entropy methods that quantify dispersion across equivalence classes of responses and are effective for hallucination detection (Farquhar et al., 2024; Kossen et al., 2024); (ii) self-consistency, which uses agreement among independently sampled answers as a proxy for confidence (Wang et al., 2023; 2025); and (iii) geometry-based measures computed from response embeddings—e.g., local density or Gram-volume statistics—that correlate with quality and robustness (Qiu & Miikkulainen, 2024; Li et al., 2025). Because these signals require neither logits nor gradients, they are natural conformity scores for our unsupervised conformal calibration; in parallel, conformal wrappers for language modeling and factuality control are emerging (Quach et al., 2024; Mohri & Hashimoto, 2024).

*Conformal prediction (CP)* is model-agnostic and supplies finite-sample, distribution-free guarantees (Angelopoulos & Bates, 2022; Vovk et al., 2005b). However, the generative workflow breaks the classical supervised setting: prompts are not quantifiable covariates. A practical procedure must therefore calibrate in a *label-only* regime and use few, parallelizable API calls to expensive models.

We introduce a practical *unsupervised conformal prediction (UCP)* framework that increases data efficiency, reduces computation via bootstrapped conformal calibration, and reconciles heterogeneous modalities through conformal alignment, while delivering distribution-free, finite-sample guarantees and strong empirical gains in hallucination detection and factuality. Our framework calibrates directly on raw outputs and is compatible with black-box APIs:

1. an LLM-friendly *atypicality* score based on inner-product interaction energy of the response-embedding Gram matrix (unit-norm, cosine), yielding a bounded, exchangeability-compatible conformity score;

2. batched unsupervised conformal procedures—split UCP for single-batch queries and new batched variants (*B-UCP* and bootstrap-stabilized *BB-UCP* )—with finite-sample coverage under batch exchangeability and improved stability/efficiency over split UCP;

3. *conformal alignment*: a batch-level calibration of a single strictness knob $\tau$ that ensures any predicate (e.g., factuality improvement) holds on unseen batches with probability at least $1 - \alpha$, enabling label-free test-time gating

The structure of the paper is outlined as follows. §2 explores query-only UQ and the foundational unsupervised CP techniques. In §3, we introduce the concepts of Gram-matrix typicality, batched/bootstrapped calibration, and conformal alignment. Experimental outcomes are presented in §4, while §5 discusses conclusions and future research directions.

## 2 BACKGROUND

### 2.1 CONFORMAL PREDICTION FOR GENERATIVE OUTPUTS

Conformal prediction (CP) is a model-agnostic, distribution-free method that turns arbitrary scores into set-valued inferences with finite-sample guarantees under a common exchangeability assumption (Vovk et al., 2005a; Angelopoulos & Bates, 2022; Lei et al., 2017). Classical assumptions can be relaxed via covariate-shift and dependence-aware extensions (Barber et al., 2023; Gibbs & Candès, 2021). Conformal Risk Control (Angelopoulos et al., 2025) not only addresses coverage but also aims to manage expected losses. Recent developments in language models use conformal calibration to enhance modeling accuracy, concentrating on removing and verifying claims (Quach et al., 2024; Mohri & Hashimoto, 2024; Cherian et al., 2024).

Building on the concepts in (Vovk et al., 2005a; Lei et al., 2017; Lei & Wasserman, 2012; Sadinle et al., 2018), we adopt the formal definitions of *Full-UCP* and *Split-UCP* as presented in Wasserman. The corresponding algorithms also appear in the Appendix. We observe the responses $Y_{1:n}$ paired with fixed prompts $X_{1:n}$ and a future pair $(X_{n+1}, Y_{n+1})$. We primarily calibrate on the responses $Y$ (i.e., model outputs), treating prompts/contexts $X$ as covariates that may be implicit. We treat $Y_{1:n}$ as exchangeable. We define residuals via a permutation-invariant map $\phi$ by $R_i = \phi(Y_i; \mathcal{S}_i)$. For tolerance $\alpha \in (0, 1)$, we target $\Pr Y_{n+1} \in C_n \geq 1 - \alpha$.

**Full-UCP** augments data with a candidate $y_{n+1}$, and residuals are recalculated for the augmented set $\{Y_1, \ldots, Y_n, y_{n+1}\}$. A conformal $p$-value is then derived from the residual ranking, incorporating $y$ when $\pi(y) \geq \alpha$. **Split-UCP** partitions the data into sets $\mathcal{D}_1 = \{Y_i : i \in I_1\}$ and $\mathcal{D}_2 = \{Y_i : i \in I_2\}$, where index sets $I_1$ and $I_2$ partition the set $\{1, \ldots, n\}$. We then calculate calibration residuals $R_i = \phi(Y_i; \mathcal{D}_1)$ for $i \in I_2$ to establish the $(1-\alpha)(1+\frac{1}{|I_2|})$-quantile $q$ of these residual values, subsequently yielding the set $C_n = \{y : \phi(y; \mathcal{D}_1) \leq q\}$. In addition, the Split-UCP marginal theorem states that under exchangeability of $(Y_{1:n}, Y_{n+1})$, the resulting set $C_n$ satisfies $\Pr\{Y_{n+1} \in C_n\} \geq 1 - \alpha$.

Full-UCP can be computationally inefficient due to retraining and ineffective searching over future candidates $y$ (grid/root-finding). On the other hand, Split-UCP is data-inefficient as only the calibration split $\mathcal{D}_2$ influences the quantile, leading to sample-splitting costs. The challenges motivate us to design our framework for better alignment with the generative stochastic process $P(Y \mid X)$ to ensure that practical applications effectively capture underlying variability.

### 2.2 GRAM MATRIX CONSTRUCTION, INNER–PRODUCT ENERGY, AND ATYPICAL SCORE

To quantify LLM response uncertainty, we build our framework on the response-embedding Gram matrix. Given $n$ responses $Y_1, \ldots, Y_n$ with embeddings $v_i := \psi(Y_i) \in \mathbb{R}^d$, we stack the embeddings as rows to create the matrix $V \in \mathbb{R}^{n \times d}$ with $V_{i,:} = v_i^\top$ and form the (uncentered) Gram $G := VV^\top \in \mathbb{R}^{n \times n}$ with entries $G_{ij} = \langle v_i, v_j \rangle$, where unit–norm embeddings $\|v_i\|_2 = 1$ by default.

We then define the inner–product (interaction) energy:

$$e(i; G) := \|G_{:,i}\|_2 = \|V v_i\|_2. \tag{2.1}$$

In unit–norm embeddings, $e(i; G) = \left( \sum_{j=1}^{n} \cos^2 \theta_{ij} \right)^{1/2}$, where $\theta_{ij}$ denotes the angle between $v_i$ and $v_j$. Thus, $e(i; G)^2$ quantifies the total squared directional alignment of $v_i$ with its peers. Since $\cos^2$ treats aligned and anti-aligned directions equally, a *large* $e(i; G)$ indicates central, redundant elements (high agreement), while a *small* $e(i; G)$ denotes unique or irrelevant content (high novelty). The following theorem states that $e(\cdot; G)$ ranges from 1 to $\sqrt{n}$ (the proof is available in the Appendix).

**Theorem 2.1** (Unit–norm interaction–energy bound). *If $\|v_i\|_2 = 1$ for all i, then $1 \leq e(i; G) \leq \sqrt{n}$ for each i. Equality $e(i; G) = \sqrt{n}$ holds when $v_i$ is perfectly aligned with all $v_j$; $e(i; G) = 1$ when $v_i$ is orthogonal to all $v_j$ for $j \neq i$.*

Letting $B_E$ denote the supremum of $e(\cdot; G)$, we define the Atypical Score as

$$\Phi(i; G) := 1 - \frac{e(i; G)}{B_E} \in [0, 1]. \tag{2.2}$$

Under unit-norm embeddings, the upper bound $B_E$ equals $\sqrt{n}$. Note that any strictly monotone transform of $e$ is equivalent for ranking.

## 3 METHODOLOGY

### 3.1 BATCH UNSUPERVISED CONFORMAL PREDICTION

We adapt UCP to a *batched* setting that gathers information across exchangeable batches and may stabilize calibration using a within-batch bootstrap. For notational simplicity, assume there are $n + 1 = (J+1)I$ responses $Y_1, \ldots, Y_{n+1}$ for pre-chosen integers $J$ and $I$. Partition the $n + 1$ responses into $J+1$ disjoint batches $\mathcal{B}_j = \{Y_{j,1}, \ldots, Y_{j,I}\}$ for $j = 1, \ldots, J+1$. The entire §3 assumes *batch exchangeability*, i.e., that the $J+1$ batches $\mathcal{B}_1, \ldots, \mathcal{B}_J, \mathcal{B}_{J+1}$ are i.i.d. and that within each batch the responses are exchangeable. (The latter condition is also called *partial exchangeability* (de Finetti, 1938).) Thus we will assume without loss of generality that $Y_{n+1} = Y_{J+1,I}$.

We design two statistically distinct methods under this setting:

- *Batch-UCP (B-UCP).* For each calibration batch $j$, residuals are computed within batches against leave-one-out Gram matrix base, where $R_{j,i} = \phi(Y_{j,i}; \mathcal{B}_{j,-i})$ using a bounded, permutation-invariant score $\phi \in [0, B_\phi]$. Pool $\{R_{j,i}\}_{j=1:J,\, i=1:I}$ and take a single adjusted conformal quantile.

- *Batch Bootstrap-UCP (BB-UCP).* For each calibration batch $j$, bootstrap the empirical residual multiset $\{R_{j,i}\}_{i=1}^{I}$ to obtain $\{S_{j,k}\}_{k=1}^{K}$. Pool $\{S_{j,k}\}_{j,k}$ and apply the same adjusted quantile. The bootstrap mitigates noise caused by irregularity from outlier batches, maintaining exchangeability.

The block below clarifies how B-UCP and BB-UCP differ. We present the formal algorithms in the Appendix (Algorithms B.3–B.4).

---

Unified Batch U-CP (demo; $K=0 \Rightarrow$ B-UCP, $K \geq 1 \Rightarrow$ BB-UCP)

1: **Inputs:** $\{Y_k\}_{k=1}^{(J+1)I-1}$, score $\phi$, batch count $J$, tolerance $\alpha$, bootstrap count $K \geq 0$
2: **Partition:** calibration $\{\mathcal{B}_j\}_{j=1}^{J}$, hold-out $\mathcal{B}_{J+1,-I}$
3: **for** $j = 1$ **to** $J$ **do**
4:     **for** $i = 1$ **to** $I$ **do** $R_{j,i} \leftarrow \phi(Y_{j,i}; \mathcal{B}_{j,-i})$, $\mathcal{B}_{j,-i} = \mathcal{B}_j \setminus \{Y_{j,i}\}$
5:     **end for**
6:     **if** $K > 0$ **then** draw $\{S_{j,k}\}_{k=1}^{K}$ from $\{R_{j,i}\}_{i=1}^{I}$
7:     **end if**
8: **end for**
9: **Bag:** $\mathcal{D} \leftarrow \{R_{j,i}\}_{j,i}$ if $K = 0$; else $\mathcal{D} \leftarrow \{S_{j,k}\}_{j,k}$
10: **Quantile:** $\delta_J = (J+1)\alpha - 1$; set $q \leftarrow B_\phi$ if $\delta_J \leq 0$, else $(1 - \delta_J/J)$-quantile of $\mathcal{D}$
11: **Output:** $C_n = \{y : \phi(y; \mathcal{B}_{J+1,-I}) \leq q\}$

---

Under the batch exchangeability assumption, we have the following coverage guarantees, and we present the proofs in the Appendix.

**Theorem 3.1** (B-UCP coverage)**.** *The prediction set $C_n$ returned by Batch U-CP satisfies* $\Pr\{Y_{n+1} \in C_n\} \geq 1 - \alpha$.

**Theorem 3.2** (BB-UCP coverage)**.** *The prediction set $C_n$ returned by Batch Bootstrap U-CP satisfies* $\Pr\{Y_{n+1} \in C_n\} \geq 1 - \alpha$.

Our design incorporates three main mechanisms.

1) *Batching.* In the unsupervised setting, conventional CP loses the easy exchangeability of supervised CP because the conformity score depends on the other responses. By organizing data into exchangeable batches and using *within-batch leave-one-out* residuals (each $R_{j,i}$ computed against a base of size $I-1$), we effectively re-enable cross-validation–style conformalization.

2) *Within-batch LOO under batch exchangeability.* This alignment makes the calibration and test residual laws match, removing the split-sample penalty in split-UCP (which inflates the order-statistic index) and yielding tighter thresholds at a fixed risk level.

3) *Bootstrap aggregation.* Averaging replicated empirical laws within each batch stabilizes the empirical quantile and down-weights idiosyncratic batches, reducing the chance of under-estimating the target quantile. Realized coverage therefore tends to be slightly conservative while intervals remain short. Resampling $\{R_{j,i}\}$ is inexpensive and preserves exchangeability; and because the method is rank-based, any strictly non-decreasing transform of $\phi$ leaves $C_n$ unchanged. These effects anticipate our observations: BB-UCP is typically more conservative than split-UCP yet produces tighter, more stable intervals.

### 3.2 CONFORMAL ALIGNMENT

Conformal alignment functions as a quality control technique across various modalities, enabling multilevel filtering and alignment superior to standard UCP schemes. Initially, we parameterize strictness using a single knob $\tau \in [0, 1]$ to filter batches with a low-cost, consistently available signal (here, the Gram matrix inner energy). During calibration, this signal is aligned with a rare or expensive quality measure (e.g., factuality), enabling deployment with just the low-cost signal while retaining the calibrated assurance $\Pr(\text{predicate on future batch}) \geq 1 - \alpha$. The idea scales to various contexts: establishing an accessible, cost-effective score allows the use of the same method to determine a global $\hat{\tau}$ from past batches, which is then applied to unlabeled new data. By utilizing text and Gram scores, the predicate can adapt any non-decreasing, right-continuous batch metric to inexpensive scores derived from text, vision, audio, or multimodal embeddings, offering substantial flexibility and a wide range of applications.

Similar to §3.1, partition the data into $J$ disjoint batches $\mathcal{B}_j = \{Y_{j,1}, \dots, Y_{j,I}\}$ for $j = 1, \dots, J$, with $\{\mathcal{B}_j\}_{j=1}^{J+1}$ exchangeable and $\mathcal{B}_{J+1}$ the future batch. Let $\mathcal{RC}$ be the space of right-continuous, non-decreasing maps $[0, 1] \to [0, 1]$, and define $\psi : \binom{\mathcal{Y}}{I} \to \mathcal{RC}$. For each batch $j$, set $\mathcal{P}_j(\cdot) = \psi(\mathcal{B}_j)$; then $\mathcal{P}_j$ is non-decreasing and right-continuous, and $\{\mathcal{P}_j\}$ is exchangeable.

We use $\mathcal{P}_j(\tau)$ as a batch predicate with a subset-selection parameter $\tau \in [0, 1]$, which describes the $j$-th batch. For instance, let $\widehat{J}_j(\tau) \subseteq \{1, \dots, I\}$ be right-continuous filtered sets; that is, for any $0 \leq \tau < \tau' \leq 1$ there exists $\delta > 0$ with $\widehat{J}_j(\tau') \subset \widehat{J}_j(\tau) = \widehat{J}_j(\tau + \delta)$. From the set $\widehat{J}_j(\tau)$ of indices, we define $\mathcal{P}_j(\tau)$ as the indicator of the event "$\widehat{J}_j(\tau)$ satisfies property A" where "property A" is to be determined according to a specific prediction target. We search for $\widehat{\tau}$ such that $\mathcal{P}_{J+1}(\widehat{\tau}) = 1$ with high probability, which means that the selected set $\widehat{J}_{J+1}(\hat{\tau})$ satisfies "property A" with high probability.

Define the minimal passing strictness

$$S_j := \min\{\tau \in [0, 1] : \mathcal{P}_j(\tau) = 1\} \in [0, 1], \tag{3.1}$$

with $\inf \varnothing = 1$. Let $K = \lceil (1 - \alpha)(J + 1) \rceil$ and calibrate $\widehat{\tau}$ as the $K$-th order statistic of $\{S_j\}_{j=1}^J$.

---

**Algorithm 3.1** Batch U-CP Conformal Alignment

---

1: **Input:** calibration batches $\mathcal{B}_1, \ldots, \mathcal{B}_J$; test batch $\mathcal{B}_{J+1,-I}$; $K \leftarrow \lceil (1-\alpha)(J+1) \rceil$; function $\psi$; tolerance $\alpha \in (0,1)$.
2: **for** $j = 1$ **to** $J$ **do**
3:      Compute $\mathcal{P}_j(\cdot) = \psi(\mathcal{B}_j)$
4:      Compute $S_j = \inf\{\tau \in [0,1]: \mathcal{P}_j(\tau) = 1\}$.
5: **end for**
6: **Calibrate** $\hat{\tau} \leftarrow$ the $K$-th smallest value among $\{S_j\}_{j=1}^J$. (If $K = J+1$, then $\hat{\tau} \leftarrow 1$)     (Split-conformal quantile with $J+1$ total batches)
7: **Output** $\hat{\tau}$.

---

A remark of Algorithm 3.1 is that, for a target threshold $r \in (0,1)$ which we want $\mathcal{P}_{J+1}(\hat{\tau}) \geq r$ to hold for high probability, define $\mathcal{P}_j'(\tau) = \mathbf{1}\{\mathcal{P}_j(\tau) \geq r\}$. Then $\mathcal{P}_j'$ is also non-decreasing and right-continuous, and $\mathcal{P}_j(\tau) \geq r \iff \mathcal{P}_j'(\tau) = 1$. Hence Algorithm 3.1 records $\{S_j'\}$ and returns a single $\hat{\tau}$ with $\Pr\{\mathcal{P}_{J+1}(\hat{\tau}) \geq r\} = \Pr\{\mathcal{P}_{J+1}'(\hat{\tau}) = 1\} \geq 1 - \alpha$ under exchangeability.

**Theorem 3.3** (B-UCP alignment guarantee). *Assume the batches $\{B_j\}_{j=1}^{J+1}$ are exchangeable (which implies that predicates $\{\mathcal{P}_j\}_{j=1}^{J+1}$ are exchangeable) and that each $\mathcal{P}_j(\cdot)$ is non-decreasing and right-continuous in its argument $\tau$ for $j = 1, \ldots, J+1$. Then Algorithm 3.1 satisfies*

$$\Pr\{\mathcal{P}_{J+1}(\hat{\tau}) = 1\} \geq 1 - \alpha.$$

By Theorem 3.3, any non-decreasing, right-continuous batch predicate calibrated across exchangeable batches yields a $1 - \alpha$ guarantee on the held-out batch. We now instantiate this scheme for black-box LLMs by selecting a cheap Gram-geometry self-consistency score $Q$ with the induced filter $\hat{J}_j(\tau)$ and a batch predicate $\mathcal{P}_j$ that encodes deployment goals (e.g., factuality lift), as specified next.

Let $Q_{j,i}$ denote the inner-product interaction energy $e(i; G)$ from Section 2.2 (unit-norm, cosine geometry). We keep high-consensus items via

$$\hat{J}_j(\tau) := \{i : Q_{j,i} > \tau\},$$

so larger $\tau$ filters out more responses.

Let $s_{j,i} \in [0,1]$ be a batch severity with larger = worse (e.g., factuality severity). In strictness $\tau$, set $K_j(\tau) = \{i : Q_{j,i} > \tau\}$ and $D_j(\tau) = \{i : Q_{j,i} \leq \tau\}$, and declare pass when the indicator is evaluated as 1, where

$$\mathcal{P}_j^{\mathrm{CVAR}}(\tau) := \mathbf{1}\Big\{ \underbrace{\mathrm{CVAR}_q(s_{j,i} : i \in D_j(\tau)) - \mathrm{CVAR}_q(s_{j,i} : i \in K_j(\tau))}_{\Delta\mathrm{CVAR}_{j,\tau}(q)} \geq \delta \Big\}.$$

CVAR focuses on the worst tail. Requiring a positive gap means that the kept set reduces severe errors (rare but damaging hallucinations) relative to the dropped set (Chow et al., 2015; Zhao et al., 2025; Rockafellar & Uryasev, 2000; Acerbi & Tasche, 2002). Because $K_j(\tau)$ enlarges as $\tau$ decreases, $\tau \mapsto \mathcal{P}_j^{\mathrm{CVAR}}(\tau)$ is non-decreasing/right-continuous, so Alg. 3.1 applies unchanged.

Let $s_{j,i}$ be a factuality severity in $[0,1]$ (lower is better; e.g., BERTScore–F1 dissimilarity). The predicate $\mathcal{P}_j^{\mathrm{F}}(\tau) = 1$ asserts that the $Q$-filtered subset attains a statistically significant median reduction in factuality severity (per the test above). Calibrating $\hat{\tau}$ across historical batches yields a single label-free gate which, using $Q$ alone at deployment, preserves this improvement on new batches with probability at least $1 - \alpha$ (Theorem 3.3). *Applications include:* (i) open-domain QA/RAG: ensure only consistent answers are shown; (ii) customer support and search snippets: reduce the risk of false confident statements; (iii) summarization/reporting: exclude sections that do not pass factual accuracy checks before publishing.

## 4 EXPERIMENTS

We perform three experiments to study three research questions aligned with §3: *(RQ1)* within a single query, does BB-UCP produce tighter prediction sets than S-UCP at the same target miscoverage?

*(RQ2)* across multiple queries, do B-/BB-UCP achieve nominal coverage and improve batch quality by discarding higher-severity responses? *(RQ3)* in cross-query alignment, can the calibrated global strictness $\hat{\tau}$ reliably yield batch-level severity reduction while preserving the conformal guarantee?

## 4.1 EXPERIMENTAL SETUP

We evaluate across four complementary QA datasets, each exhibiting a distinct failure mode: *ASQA*—ambiguity and underspecification (Stelmakh et al., 2023); *NQ-Open*—single-hop factoid retrieval (Lee et al., 2019; Kwiatkowski et al., 2019); *HotpotQA*—multi-hop composition (Yang et al., 2018); and *AmbigQA*—aliases and answer sets (Min et al., 2020). To probe sensitivity, we add two ablations: a decoding-entropy stress test and a vendor/model swap. For each open-domain QA prompt, we (i) synthesize a diverse response set by mixing plain answers, lightly *enforced* canonical answers, and structured *noise* outliers. This controlled injection is standard for stress test hallucination detection and semantic dispersion UQ signals (Kuhn et al., 2023; Qiu & Miikkulainen, 2024) and allows us to probe robustness under realistic contamination. All texts are embedded with a lightweight sentence encoder (`all-MiniLM-L6-v2`). We stack unit-normalized embedding vectors by rows to form $V$ and then the Gram matrix $G = VV^{\top}$; (ii) expand the reference set with concise paraphrases to reduce aliasing; and (iii) characterize each candidate using a distance-based metric, **Factuality Severity** (FS). All artifacts are logged and kept provider-agnostic across OpenAI, Together, and Gemini (OpenAI et al., 2024; Grattafiori et al., 2024; Team et al., 2024).

We quantify answer quality with respect to references using BERTScore–F1 with baseline rescaling (`roberta-large`) (Zhang et al., 2020) on the *answer head* (first sentence or a `Final:` field, truncated to $\leq 16$ tokens). Letting $\text{head}(a)$ be the head of answer $a$ and $\mathcal{R}_q$ the reference set, we define the severity

$$\text{FS}(a) := 1 - \max_{r \in \mathcal{R}_q} \text{BERTScoreF1}\big(\text{head}(a), r\big) \in [0, 1], \qquad (4.1)$$

so that a value of 0 indicates a near-paraphrase of some reference (high factual alignment), and a value near 1 flags semantic deviation. Scoring the head avoids rationale contamination and normalizes across style/length.

Given a batch $B$ and the kept subset $K(\tau)$ after filtering by $Q$, we summarize factuality lifting by the *median reduction*

$$\Delta_{\text{FS}}(\tau) = \text{median}\{FS(y) : y \in B\} - \text{median}\{FS(y) : y \in K(\tau)\}$$

(larger is better). All conformal calibrations use the Gram inner–energy score $e$ from §3 and are implemented exactly as specified (*S-/B-/BB-UCP* and alignment). We fix random seeds, cache embeddings/Grams, and run identical pipelines on all datasets.

## 4.2 EXPERIMENT I — SINGLE-QUERY CONFORMAL CALIBRATION (S-UCP VS. BB-UCP)

In the single-query regime, for each question $q$ we embed all candidate responses, form the unit-norm response Gram matrix $G$, and compute the inner-product energy score $Q$ for each response. We then construct residuals within the same pool and repeatedly split into calibration/test subsets of fixed sizes. Split UCP thresholds test residuals by directly taking a quantile of residuals; BB-UCP additionally bootstraps the calibration residuals within the batch and aggregates the quantiles to stabilize the threshold. We report (i) empirical coverage against the $1 - \alpha$ target and (ii) an efficiency proxy given by the accepted sublevel endpoint $q_{1-\alpha}$ (smaller is better).

Across *AmbigQA*, *AmbigQA-ENT*, *ASQA*, and *HotpotQA*, both S-UCP and BB-UCP achieve near-nominal coverage, while BB-UCP consistently achieves more conservative empirical coverage across repetitions and $\alpha$ and yields shorter interval length; see Fig. 1 (top/bottom). On *NQ-Open* and *NQ-Open-Vend*, performance is weaker: the answer pools are small and low-diversity, often collapsing into a single high-consensus "heap." This compresses dispersion in Gram space, produces near-tied residual ranks, and blunts the bootstrap's advantage; Split UCP 's coverage remains close to target. Slightly enlarging pool size or boosting response diversity effectively restores similar qualitative improvements as seen in other datasets.

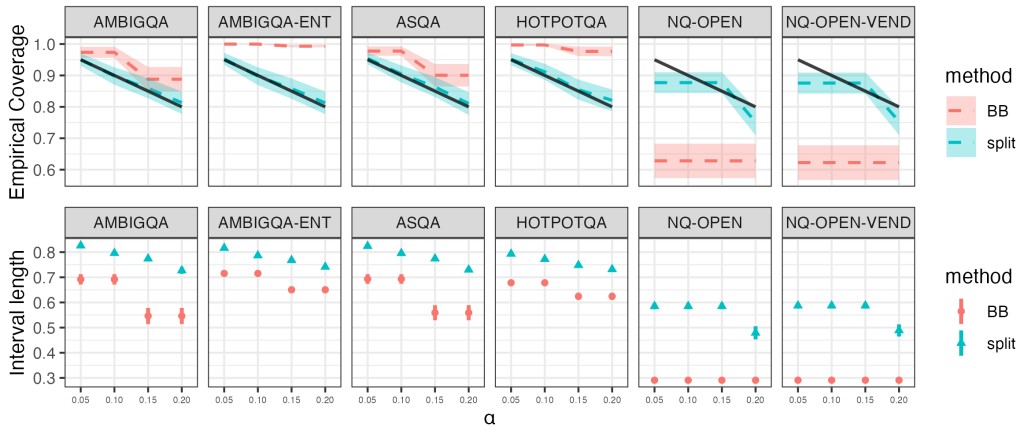

Figure 1: **Experiment 1. Top row:** Empirical coverage vs. $\alpha$ for a representative dataset (ASQA). Shaded bands show $\pm 1$ SE, and the black line is the $1-\alpha$ target. Both the BB and Split UCP achieve the desired theoretical assurance, consistently staying above the target. BB is typically more conservative to attain greater empirical coverage. **Bottom row:** Interval size comparison bar are plotted across datasets, grouped by method (BB, split) and stratified by $\alpha \in \{0.05, 0.10, 0.15, 0.20\}$. In *all* datasets and at *all* $\alpha$ levels, BB consistently produces smaller interval sizes than Split, indicating the bootstrapping achieves statistical efficiency as intended.

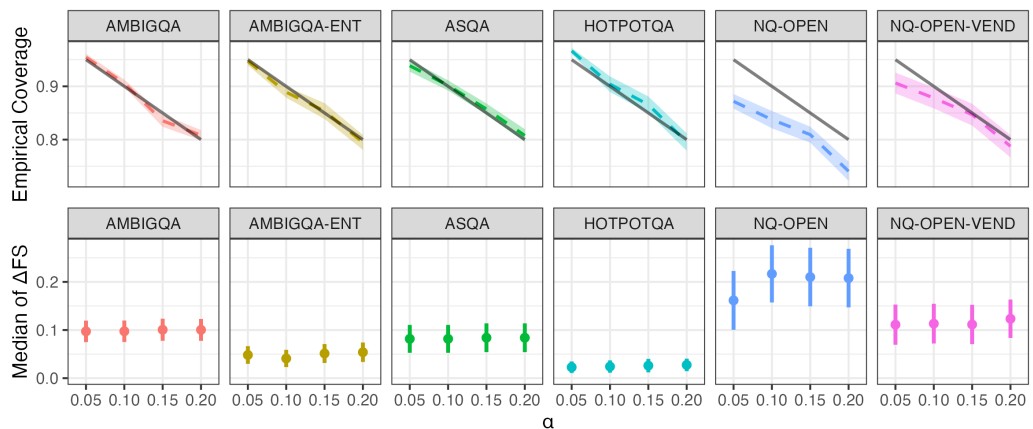

Figure 2: **Experiment 2. Top row:** Empirical coverage vs. $\alpha$. Each panel corresponds to one dataset. Shaded bands show $\pm$ SE, and the black line is the $1-\alpha$ target. Across all datasets, cross-query BB-UCP nearly achieves the expected theoretical guarantee, with a single failure occurring solely in an extreme stress test. **Bottom row:** The change in median factuality severity ($\Delta$FS) is shown across datasets (points indicate the mean, intervals represent $\pm$ SE). For every dataset, $\Delta$FS constantly stay above 0, highlighting the efficacy of BB-UCP in improving factuality.

### 4.3    EXPERIMENT II — CROSS-QUERY CALIBRATION (BB-UCP) AND FACTUALITY LIFTING

To mimic real-world deployment, each response set of the same query forms a *batch*. We run leave-one-query-out (LOQO) cross-validation: hold one batch for testing and calibrate on the rest. For every calibration batch we compute residuals from $Q$ under within-batch LOO, bootstrap residuals per batch, pool the bootstrapped $(1-\alpha)$-quantiles into a single threshold $q_{1-\alpha}^{\text{BB}}$, and apply that global threshold to the held-out batch. We evaluate (i) empirical coverage vs. $1-\alpha$ and (ii) factuality lifting via the median reduction in factuality severity, $\Delta$FS (excluded minus kept; better positivity indicates improved prevention of hallucinations.).

LOQO empirical coverage closely tracks $1 - \alpha$ across risk levels and datasets (Fig. 2, top), demonstrating that a *BB-UCP* global threshold learned from historical queries generalizes to unseen queries. More importantly for deployment, filtering by $q_{1-\alpha}^{\mathrm{BB}}$ consistently improves batch quality: $\Delta$FS $> 0$ across *all* datasets and *all* $\alpha$ (Fig. 2, bottom; The hardest panels are instructive: on *NQ-Open*, median $\Delta$FS $\approx 0.209$ (largest among benchmarks) despite average empirical coverage falling short of the $95\%$ target ($88.98\%$ vs. $95.00\%$); *NQ-Open-Vend* shows the same trade-off (median $\Delta$FS $\approx 0.112$, $92.88\%$ vs. $95.00\%$) (Appendix C, Table 4). This under-coverage is attributable to small-$N$/low-entropy pools that yield discretization effects and near-ties in residual ranks. As in Experiment I, standard operational tweaks—increasing per-query pool size or adding response diversity—tighten coverage without erasing the observed factuality lift.

### 4.4 EXPERIMENT III — CROSS-QUERY CONFORMAL ALIGNMENT (CVAR-GAP)

We perform LOQO folds for conformal alignment as well. In each fold, for every calibration batch $j$ we scan a strictness grid $\tau$ and evaluate the CVaR-gap predicate $\mathcal{P}_j^{\mathrm{CVAR}}(\tau)$ at tail level $q$ and margin $\delta$, then record the minimal passing strictness $S_j$. We conformally calibrate a global $\hat{\tau}$ as the split-batch $(1 - \alpha)$-quantile of $\{S_j\}$ (Alg. 3.1), then deploy on the held-out batch using only the cheap score $Q$: keep $K(\hat{\tau}) = \{i : Q_i > \hat{\tau}\}$ and report (i) empirical pass rate against $1 - \alpha$ and (ii) factuality improvement via $\Delta\mathrm{CVAR}(\hat{\tau})$. Full predicate and implementation details are in Appendix B.5/C.

Alignment preserves the statistical target while delivering consistent factuality gains: for every dataset and every risk level, the reduction in factuality severity on the kept set is positive on average (Fig. 3). Notably, the largest median and mean improvements occur on the two hardest datasets—*NQ-Open* and *NQ-Open-Vend*—with median $\Delta$FS $\approx 0.206$ and $\approx 0.112$, respectively (Appendix C, Table 5). Aligning the affordable Gram-geometry score with the factuality signal provides an effective, economical filtering method, ensuring conformal assurance for new batches.

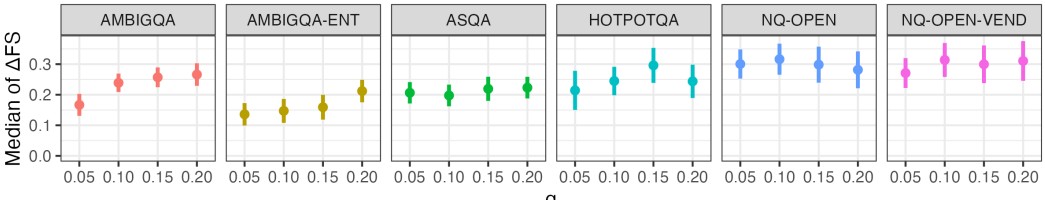

Figure 3: **Experiment 3.** Median of $\Delta$FS by dataset (points: mean, intervals: $\pm$ SE), and each panel corresponds to one dataset. Across *every* dataset and at *every* risk level, the reduction in factuality severity is consistently above 0 on average, demonstrating that conformal alignment with the gram matrix energy effectively enhances factuality while preserving statistical guarantees.

## 5 CONCLUSION

We introduced an unsupervised conformal inference framework for black-box LLMs that operates entirely on sampled outputs. The framework comprises three deployable components: (i) a Gram-geometry *atypicality* score based on unit-norm response-embedding inner products, yielding a bounded, interpretable, and stable signal; (ii) batched conformal procedures (B-UCP and the bootstrap-stabilized BB-UCP) that provide distribution-free, finite-sample guarantees under batch exchangeability while improving quantile stability and data efficiency over split UCP; and (iii) *conformal alignment*, which calibrates a global strictness parameter $\hat{\tau}$ so that a batch predicate (e.g., a CVaR-gap factuality lift) holds on new batches with probability at least $1 - \alpha$. Conformal alignment provides a principled way to synchronize an expensive signal (ground truth) with a cheaper proxy and performs well when deployment relies only on the proxy, yielding a probabilistic approach to multimodal signal gating and filtering.

**Limitations and outlook.** Our guarantees assume exchangeability across batches and within-batch permutation invariance; violations due to drift or covariate shift motivate weighted or

covariate-aware variants. Performance depends on embedding quality and normalization, underscoring the need for robustness audits and principled model/embedding selection. Predicate design (e.g., CVaR vs. median lifts and multi-metric trade-offs) invites cost-aware utilities and multi-task calibration. Extending alignment to multimodal settings and adding adaptive or online recalibration are promising directions for stronger reliability under non-stationarity.

**Use of AI for language editing.** We used OpenAI ChatGPT and Overleaf Writefull solely for language polishing (grammar, clarity, style) of author-written text. All ideas, experiments, and conclusions are the authors' own, and the authors reviewed and take responsibility for all content.

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

# A PROOFS

## A.1 GRAM MATRIX

**Theorem 2.1** (Unit–norm interaction–energy bound). *If $\|v_i\|_2 = 1$ for all $i$, then $1 \le e(i;G) \le \sqrt{n}$ for each $i$. Equality $e(i;G) = \sqrt{n}$ holds when $v_i$ is perfectly aligned with all $v_j$; $e(i;G) = 1$ when $v_i$ is orthogonal to all $v_j$ for $j \ne i$.*

*Proof.*

$$e(i;G) := \|G_{:,i}\|_2 = \Big(\sum_{j=1}^{n}\langle v_i, v_j\rangle^2\Big)^{1/2} = \Big(\|v_i\|^2 + \sum_{j \ne i}^{n}\langle v_i, v_j\rangle^2\Big)^{1/2}$$

Since $\|v_i\|_2 = 1$, the $j = i$ term equals 1, so

$$e(i;G)^2 = 1 + \sum_{j \ne i}\langle v_i, v_j\rangle^2.$$

Now for $j \ne i$, because both $v_i$ and $v_j$ are unit vectors, $\langle v_i, v_j\rangle = \cos\theta_{ij} \in [0,1]$. Therefore,

$$1 \le e(i;G)^2 \le 1 + (n-1) = n,$$

and taking square roots gives the bound

$$1 \le e(i;G) \le \sqrt{n}.$$

The lower bound is attained when $v_i$ is orthogonal to all $v_j$'s with $j \ne i$; the upper bound is attained when $v_i$ is perfectly aligned or anti-aligned with all $v_j$'s. $\qquad\square$

## A.2 SPLIT UCP

**Theorem A.1** (Split-UCP marginal coverage). *For exchangeable $\{Y_i\}_{i=1}^{n+1}$ and $\phi$, the split conformal prediction set as stated in Section 3 satisfies $\Pr\{Y_{n+1} \in C_n\} \ge 1 - \alpha$.*

*Proof.* Let $R_{n+1} = \phi(Y_{n+1}; \mathcal{D}_1)$. Given $\mathcal{D}_1$, the distribution of $(R_i)_{i \in I_2 \cup \{n+1\}}$ is exchangeable. Hence

$$\Pr\{Y_{n+1} \in C_n\} = \Pr\Big\{R_{n+1} \le \text{ the } \lceil(1-\alpha)(|I_2|+1)\rceil\text{-th smallest of } \{R_i\}_{i \in I_2}\Big\}$$

$$= \Pr\Big\{R_{n+1} \le \text{ the } \lceil(1-\alpha)(|I_2|+1)\rceil\text{-th smallest of } \{R_i\}_{i \in I_2 \cup \{n+1\}}\Big\}$$

$$= \frac{\lceil(1-\alpha)(|I_2|+1)\rceil}{|I_2|+1} \ge 1-\alpha.$$

$$\square$$

## A.3 BATCH U-CP

**Theorem 3.1** (B-UCP coverage). *The prediction set $C_n$ returned by Batch U-CP satisfies $\Pr\{Y_{n+1} \in C_n\} \ge 1 - \alpha$.*

*Proof.* If $\frac{1}{J+1} \ge \alpha$, then $C_n$ is the whole space and the claim is trivial. Now assume $\frac{1}{J+1} < \alpha$. Define the (random) indicator function

$$L_{j,i}(\lambda) := \mathbf{1}\big\{\phi(Y_{j,i}, \mathcal{B}_{j,-i}) > \lambda\big\}, \qquad j = 1, \dots, J+1, \ i = 1, \dots, I.$$

Since $Y_{J+1,I}$ is not used to construct $C_n$, we have

$$\Pr(Y_{n+1} \in C_n) = \Pr\big(\phi(Y_{n+1}, \mathcal{B}_{J+1,-I}) \le q\big) = 1 - \mathbb{E}[L_{J+1,I}(q)],$$

where $q$ is the $\big(1 - \frac{(J+1)\alpha - 1}{J}\big)$-quantile of the set $\{\phi(Y_{j,i}, \mathcal{B}_{j,-i})\}_{j=1:J, i=1:I}$. Define

$$\hat{\lambda}' := \inf\left\{\lambda: \frac{1}{(J+1)I}\sum_{j=1}^{J+1}\sum_{i=1}^{I} L_{j,i}(\lambda) \le \alpha\right\}, \quad \hat{\lambda} := \inf\left\{\lambda: \frac{1}{(J+1)I}\sum_{j=1}^{J}\sum_{i=1}^{I} L_{j,i}(\lambda) + \frac{1}{J+1} \le \alpha\right\}.$$

These two $\lambda$'s exist because $L_{j,i}(B_\phi) = 0$ for any $(i, j)$.

Then $\hat{\lambda}' \leq \hat{\lambda} \leq q$, so $L_{J+1,i}(\hat{\lambda}') \geq L_{J+1,i}(\hat{\lambda}) \geq L_{J+1,i}(q)$ for all $i$. Therefore

$$
\mathbb{E}[L_{J+1,I}(q)] \leq \mathbb{E}\left[L_{J+1,I}(\hat{\lambda})\right] = \mathbb{E}\left[\mathbb{E}\left[L_{J+1,I}(\hat{\lambda}) \mid \{\mathcal{B}_j\}_{j=1}^J\right]\right] = \mathbb{E}\left[\tfrac{1}{I}\sum_{i=1}^I L_{J+1,i}(\hat{\lambda})\right]
$$

where the last equality uses exchangeability within batch $B_{J+1}$. Because $\hat{\lambda}' \leq \hat{\lambda}$, the right-most term is bounded above by

$$
\mathbb{E}\left[\tfrac{1}{I}\sum_{i=1}^I L_{J+1,i}(\hat{\lambda}')\right] = \mathbb{E}\left[\tfrac{1}{(J+1)I}\sum_{j=1}^{J+1}\sum_{i=1}^I L_{j,i}(\hat{\lambda}')\right] \leq \mathbb{E}\left[\tfrac{1}{(J+1)I}\sum_{j=1}^{J+1}\sum_{i=1}^I L_{j,i}(\hat{\lambda}')\right] \leq \alpha
$$

by also using exchangeability of the batches $\{\mathcal{B}_j\}_{j=1}^{J+1}$, the inequality $\hat{\lambda} \leq q$, and then the definition of $q$. Thus $\Pr(Y_{n+1} \in C_n) \geq 1 - \alpha$. $\square$

## A.4 BATCH BOOTSTRAP U-CP

**Theorem 3.2** (BB-UCP coverage). *The prediction set $C_n$ returned by Batch Bootstrap U-CP satisfies $\Pr\{Y_{n+1} \in C_n\} \geq 1 - \alpha$.*

*Proof.* As with the proof in Section A.3, we have

$$
\Pr(Y_{n+1} \in C_n) = \Pr(\phi(Y_{n+1}, \mathcal{B}_{J+1,-I}) \leq q) = 1 - \mathbb{E}[\mathbf{1}\{\phi(Y_{n+1}, \mathcal{B}_{J+1,-I}) > q\}].
$$

Also do virtual bootstrap in the future batch $\mathcal{B}_{J+1}$ to get $\{S_{J+1,k}\}_{k=1}^K$. Independence across $j$ and identical distributions imply

$$
\mathbb{E}[\mathbf{1}\{\phi(Y_{n+1}, \mathcal{B}_{J+1,-I}) > q\}] = \mathbb{E}\left[\tfrac{1}{K}\sum_{k=1}^K \mathbf{1}\{S_{J+1,k} > q\}\right].
$$

Let

$$
\hat{\lambda}' := \inf\left\{\lambda : \frac{1}{(J+1)K}\sum_{j=1}^{J+1}\sum_{k=1}^K \mathbf{1}\{S_{j,k} > \lambda\} \leq \alpha\right\}
$$

$$
\hat{\lambda} := B_\phi \wedge \inf\left\{\lambda : \frac{1}{(J+1)K}\sum_{j=1}^J\sum_{k=1}^K \mathbf{1}\{S_{j,k} > \lambda\} + \frac{1}{J+1} \leq \alpha\right\}.
$$

Then $\hat{\lambda}' \leq \hat{\lambda} \leq q$ and the same exchangeability argument yields

$$
\mathbb{E}[\mathbf{1}\{\phi(Y_{n+1}, \mathcal{B}_{J+1,-I}) > q\}] = \mathbb{E}\left[\tfrac{1}{K}\sum_{k=1}^K \mathbf{1}\{S_{J+1,k} > q\}\right]
$$

$$
\leq \mathbb{E}\left[\tfrac{1}{K}\sum_{k=1}^K \mathbf{1}\{S_{J+1,k} > \hat{\lambda}'\}\right]
$$

$$
= \mathbb{E}\left[\tfrac{1}{(J+1)K}\sum_{j=1}^{J+1}\sum_{k=1}^K \mathbf{1}\{S_{j,k} > \hat{\lambda}'\}\right]
$$

$$
\leq \alpha,
$$

hence $\Pr(Y_{n+1} \in C_n) \geq 1 - \alpha$. $\square$

## A.5 BATCH-WISE CONFORMAL ALIGNMENT

**Theorem 3.3** (B-UCP alignment guarantee). *Assume the batches $\{B_j\}_{j=1}^{J+1}$ are exchangeable (which implies that predicates $\{\mathcal{P}_j\}_{j=1}^{J+1}$ are exchangeable) and that each $\mathcal{P}_j(\cdot)$ is non-decreasing and right-continuous in its argument $\tau$ for $j = 1, \ldots, J+1$. Then Algorithm 3.1 satisfies*

$$\Pr\{\ \mathcal{P}_{J+1}(\hat{\tau}) = 1\ \} \ \geq\ 1 - \alpha.$$

*Proof.* The algorithm statement provides strictness $S_j := \inf\{\tau : \mathcal{P}_j(\tau) = 1\}$ with $\inf \varnothing = 1$, value $K = \lceil (1 - \alpha)(J + 1) \rceil$, and $\hat{\tau}$ being the $K$-th order statistic of $\{S_j\}_{j=1}^{J}$. Let $\hat{\tau}_{J+1}$ be the $K$-th order statistic of $\{S_j\}_{j=1}^{J+1}$. Noting that exchangeability of $\{\mathcal{P}_j\}_{j=1}^{J+1}$ implies exchangeability of $\{S_j\}_{j=1}^{J+1}$, we get

$$\Pr\{\mathcal{P}_{J+1}(\hat{\tau}) = 1\} = \Pr\{S_{J+1} \leq \hat{\tau}\} = \Pr\{S_{J+1} \leq \hat{\tau}_{J+1}\} = \frac{K}{J+1} \geq 1 - \alpha.$$

$\square$

*Remark.* The proof skills for Theorem 3.1 and 3.2 comes from the attempt to go back from conformal risk control to conformal prediction. In the proof we constructed a proper loss function to achieve this with a standard argument of CRC. This technique can be transfer to many other settings where we need to go back to CP from CRC.

## B ALGORITHMS

### B.1 CLASSICAL FULL UCP (BACKGROUND)

---
**Algorithm B.1** Full Unsupervised Conformal Prediction (Full-UCP)

---
1: **Input:** Data $Y_{1:n}$, score function $\phi$, tolerance $\alpha \in (0, 1)$
2: **Output:** prediction set $C_n$
3: **for** candidate $y \in \mathbb{R}$ (grid or root-finding) **do**
4:      Form $\mathcal{A} \leftarrow \{Y_1, \ldots, Y_n, y\}$
5:      Compute residuals $R_i \leftarrow \phi(Y_i; \mathcal{S}_i)$ for $i = 1, \ldots, n$, and $R_{n+1} \leftarrow \phi(y; \mathcal{S}_{n+1})$
6:      Compute the $p$-value $\pi(y) \leftarrow \dfrac{1}{n+1} \sum_{i=1}^{n+1} \mathbf{1}\{R_i \geq R_{n+1}\}$
7: **end for**
8: **Return** $C_n \leftarrow \{y : \pi(y) \geq \alpha\}$

---

### B.2 SPLIT UCP (BACKGROUND)

---
**Algorithm B.2** Split Unsupervised Conformal Prediction (Split-U-CP)

---
1: **Input:** $Y_{1:n}$, score $\phi$, tolerance $\alpha$
2: Randomly form $I_1, I_2$; define $\mathcal{D}_1$
3: Compute residuals $R_i \leftarrow \phi(Y_i; \mathcal{D}_1)$ for $i \in I_2$
4: $q \leftarrow \mathcal{Q}_{(1-\alpha)(1+\frac{1}{|I_2|})}(\{R_i : i \in I_2\})$
5: **Return** $C_n \leftarrow \{y : \phi(y; \mathcal{D}_1) \leq q\}$.

---

### B.3 Batch UCP and Batch Bootstrap U-CP (Formal Algorithms)

---

**Algorithm B.3** Batch U-CP

---

1: **Input:** responses $\{Y_k\}_{k=1}^{(J+1)I-1}$, score $\phi$ (bounded by $B_\phi$), batch count $J$, tolerance $\alpha$.
2: **Partition** the data into $J+1$ disjoint batches $B_j = \{Y_{j,1}, \ldots, Y_{j,I}\}$ for $j = 1, \ldots, J$ and $B_{J+1,-I} = \{Y_{J+1,1}, \ldots, Y_{J+1,I-1}\}$.
3: **for** $j = 1$ **to** $J$ **do**                    ▷ Within-batch leave-one-out residuals
4:     **for** $i = 1$ **to** $I$ **do**
5:         $R_{j,i} \leftarrow \phi(Y_{j,i}; B_{j,-i})$ where $B_{j,-i} = B_j \setminus \{Y_{j,i}\}$
6:     **end for**
7: **end for**
8: **if** $\frac{1}{J+1} \le \alpha$ **then**
9:     $q \leftarrow \big(1 - \frac{(J+1)\alpha-1}{J}\big)$-quantile of $\{R_{j,i}\}_{j,i}$
10: **else**
11:     $q \leftarrow B_\phi$
12: **end if**
13: **Return** $C_n = \big\{y : \phi(y; B_{J+1,-I}) \le q\big\}$

---

---

**Algorithm B.4** Batch Bootstrap U-CP

---

1: **Input:** responses $\{Y_k\}_{k=1}^{(J+1)I-1}$, score $\phi$ (bounded by $B_\phi$), batch count $J$, tolerance $\alpha$, bootstrap count $K$.
2: **Partition** as in Algorithm B.3.
3: **for** $j = 1$ **to** $J$ **do**                    ▷ Within-batch leave-one-out residuals
4:     **for** $i = 1$ **to** $I$ **do**
5:         $R_{j,i} \leftarrow \phi(Y_{j,i}; B_{j,-i})$
6:     **end for**
7:     Draw $K$ bootstrap replicates $\{S_{j,\ell}\}_{\ell=1}^K$ from $\{R_{j,i}\}_{i=1}^I$
8: **end for**
9: **if** $\frac{1}{J+1} \le \alpha$ **then**
10:     $q \leftarrow \big(1 - \frac{(J+1)\alpha-1}{J}\big)$-quantile of $\{S_{j,\ell}\}_{j,\ell}$
11: **else**
12:     $q \leftarrow B_\phi$
13: **end if**
14: **Return** $C_n = \big\{y : \phi(y; B_{J+1,-I}) \le q\big\}$

---

### B.4 Algorithmic details for the CVaR-gap predicate

Let $Q_{j,i}$ denote the inner-product interaction energy from Section 2.2 (unit-norm, cosine geometry). We keep high-consensus items via

$$\widehat{J}_j(\tau) := \{i : Q_{j,i} > \tau\},$$

so larger $\tau$ retains fewer and more self-consistent responses.

Let $s_{j,i} \in [0,1]$ be a batch severity with larger = worse (e.g., factuality severity). At strictness $\tau$, define the kept and dropped sets $K_j(\tau) = \{i : Q_{j,i} > \tau\}$ and $D_j(\tau) = \{i : Q_{j,i} \le \tau\}$. For a random variable $X$ with CDF $F_X$, the upper-tail *Conditional Value-at-Risk* at level $q \in (0,1)$ is

$$\mathrm{CVAR}_q(X) := \frac{1}{1-q} \int_q^1 \mathrm{VaR}_u(X)\, du, \quad \text{where } \mathrm{VaR}_u(X) = \inf\{x : F_X(x) \ge u\}.$$

We instantiate a batch predicate that asks for a *tail-risk improvement* after filtering:

$$\Delta\mathrm{CVAR}_{j,\tau}(q) := \mathrm{CVAR}_q\big(s_{j,i} : i \in D_j(\tau)\big) - \mathrm{CVAR}_q\big(s_{j,i} : i \in K_j(\tau)\big),$$

$$\mathcal{P}_j^{\mathrm{CVAR}}(\tau) := \mathbf{1}\{\Delta\mathrm{CVAR}_{j,\tau}(q) \ge \delta\}.$$

| ID | Benchmark (split) | #Q | Para $P$ | Ans $N$ | Mix (N/E/Z) | Entropy $\tau$ |
|----|-------------------|-----|----------|---------|-------------|----------------|
| C1 | ASQA (dev) | 60 | 10 | 150 | (.75/.00/.25) | 0.90 |
| C2 | NQ-Open (val) | 60 | 6 | 16 | (.67/.00/.33) | 0.86 |
| C3 | HotpotQA (val) | 60 | 10 | 100 | (.60/.00/.40) | 0.86 |
| C4 | AmbigQA (dev) | 60 | 10 | 150 | (.75/.00/.25) | 0.86 |
| C5 | AmbigQA (dev) (ablation: decoding entropy) | 40 | 10 | 150 | (.75/.00/.25) | 0.86 |
| C6 | NQ-Open (val) (ablation: vendor/model) | 60 | 6 | 16 | (.67/.00/.33) | 0.86 |

Table 1: Benchmarks and per-item sampling settings used in the hallucination study. The mix column shows (normal/enforced/noise).

CVAR, also known as *Expected Shortfall*, focuses on the worst tail and is coherent and robust to heavy tails; demanding a positive CVAR gap concentrates the kept set on reliably low-severity answers and suppresses rare but severe failures.

---

**Algorithm B.5** CVaR-gap alignment: minimal strictness and split-batch calibration

---

1: **Inputs:** calibration batches $\{B_j\}_{j=1}^J$; held-out test batch $B_{J+1}$; Gram score $Q \in [0,1]$; tail level $q \in (0,1)$; margin $\delta \geq 0$; miscoverage $\alpha \in (0,1)$.
2: **Kept/excluded at strictness** $\tau$: $K_j(\tau) = \{i : Q_{j,i} > \tau\}$, $E_j(\tau) = B_j \setminus K_j(\tau)$.
3: **Empirical** $\text{CVaR}_q$. For a multiset $S \subset [0,1]$ with $m = |S|$, sort in descending order $s_{(1)} \geq \cdots \geq s_{(m)}$ and set $h = \lceil (1-q)m \rceil$, $\widehat{\text{CVaR}}_q(S) = \frac{1}{h} \sum_{\ell=1}^h s_{(\ell)}$ (winsorize if $h=0$).
4: **Predicate:** $\mathcal{P}_j^{\text{CVaR}}(\tau) = \mathbf{1}\{ \widehat{\text{CVaR}}_q(s|E_j(\tau)) - \widehat{\text{CVaR}}_q(s|K_j(\tau)) \geq \delta \}$.
5: **Minimal strictness (per batch).** Scan $\tau$ over the right-continuous grid induced by the unique $Q$ values in $B_j$ (plus 0 and 1). Let $S_j = \inf\{\tau \in [0,1] : \mathcal{P}_j^{\text{CVaR}}(\tau) = 1\}$ (set $S_j \leftarrow 1$ if the set is empty).
6: **Split-batch calibration.** Return $\widehat{\tau} = Quant_{1-\alpha}(\{S_j\}_{j=1}^J)$ as in §3.2.
7: **Deployment on** $B_{J+1}$. Keep $K_{J+1}(\widehat{\tau}) = \{i : Q_{J+1,i} > \widehat{\tau}\}$ and report the CVaR-gap.

---

Let $FS_{j,i}$ be a factuality severity in $[0,1]$ (lower is better; e.g., BERTScore–F1 dissimilarity). The predicate $\mathcal{P}_j^{\text{F}}(\tau) = 1$ asserts that the $Q$-filtered subset achieves a statistically significant median reduction in factuality severity (per the test above). Calibrating $\hat{\tau}$ across historical batches yields a single label-free gate which, when applied with $Q$ alone, preserves this improvement on new batches with probability at least $1 - \alpha$ (Theorem 3.3).

## C EXPERIMENT

### C.1 APPENDIX: HALLUCINATION EXPERIMENT SETTINGS AND CONFIGURATIONS

For each question we generate a response set, compute **Factuality Severity** $= 1 - \max_{r \in \text{refs}} \text{BERTScore-}F1(a,r)$. All runs are seeded and logged to timestamped, self-describing CSVs: a per-*answer* file (scores, margins, types, decoding knobs) and a per-*run* file (dataset/split, sample counts, model/provider, seeds, thresholds, and paths). Together model IDs are normalized to serverless fallbacks to avoid availability regressions.

We evaluate across four core datasets—ASQA (dev), NQ-Open (validation), HotpotQA (validation), AmbigQA (dev)—plus two ablations that stress decoding entropy and vendor/model choice. Each configuration fixes decoding knobs and the normal/enforced/noise mix, while paraphrasing a canonical gold to reduce aliasing of surface forms.

*Shared knobs:* alias-normalization for Together; n_per_call=5; rate-limit $\approx$ 0.8s; severity mix weight logged; seeds: C1=42, C2=7, C3=11, C4=23, C5=23, C6=8.

We use minimal, auditable prompts. For *paraphrasing* the canonical gold: *System:* "You rewrite text. Output a succinct standalone paraphrase." *User:* "Paraphrase the following answer in different wording, preserving the exact meaning and factual content. Keep it concise and standalone. Avoid hedging, qualifiers, or extra details. **Answer:** {*gold*}." For *normal answers*: *System:* "Answer the question with the canonical short answer

| ID | Provider | Model | Temp | Top-$p$ | MaxTok | Embed | BERTScore |
|----|----------|-------|------|---------|--------|-------|-----------|
| C1 | Together | Llama-3.3-70B-Instr. Turbo | 1.3 | 1.0 | 256 | MiniLM-L6-v2 | RoBERTa-large |
| C2 | OpenAI | gpt-4o-mini | 0.1 | 1.0 | 96 | MiniLM-L6-v2 | RoBERTa-large |
| C3 | Together | Mixtral-8x7B-Instr. v0.3 | 1.2 | 1.0 | 256 | MiniLM-L6-v2 | RoBERTa-large |
| C4 | Together | Llama-3.1-8B-Instr. Turbo | 0.7 | 0.9 | 256 | MiniLM-L6-v2 | RoBERTa-large |
| C5 | Together | Llama-3.1-8B-Instr. Turbo | 1.3 | 1.0 | 256 | MiniLM-L6-v2 | RoBERTa-large |
| C6 | Together | Llama-3.1-8B-Instr. Turbo | 0.1 | 1.0 | 96 | MiniLM-L6-v2 | RoBERTa-large |

Table 2: Provider/decoding and measurement settings, linked by **ID** to Table 1.

first; then add at most one brief justification. Be concise." *User:* {*question*}. For *enforced canonical answers*: *System:* "Answer with the canonical short answer first; then a single, concrete supporting detail. Avoid aliasing, avoid hedging, avoid contradictory statements." *User:* {*question*}. (Noise/outlier strings are programmatically injected: gibberish, off-topic, fabricated citations, prompt-injection strings, contradictions, emoji floods, and multilingual snippets.)

Embeddings: `sentence-transformers/all-MiniLM-L6-v2` with unit-norm rows; semantic-entropy uses a soft neighbor kernel above $\tau$ (exponent $\kappa=4$) and a normalized $-\log$ mapping to $[0, 1]$. Severity-F1 uses `bert-score` with `roberta-large` (baseline-rescaled) on "answer heads" (first $\leq 16$ tokens) to limit verbosity bias. All artifacts are timestamped and saved as {`dataset`}_{`model`}_{`stamp`}_ns{`N`} for direct reuse in downstream risk control. This mirrors the same compute-aware calibration-to-deployment recipe we use for LLM-as-Judge.

## C.2 MORE RESULTS

Table 3: **Benchmark mapping used in the six-panel comparisons for plotting.** Short codes are the compact labels used in figure titles. For JudgeQ CSVs, the same names appear with the suffix `__judged`.

| Panel | Short code | CSV `dataset_name` |
|-------|-----------|---------------------|
| 1 | AMBIGQA-ENT | `ambigqa__llama8b__hiT__ablation_entropy__ns40_responses` |
| 2 | AMBIGQA | `ambigqa__llama8b__midT_ns60_responses` |
| 3 | ASQA | `asqa__llama70b__hiT_ns60_responses` |
| 4 | HOTPOTQA | `hotpot_mixtral8x7b__hiT_noise40_ns60_responses` |
| 5 | NQ-OPEN | `nq__gpt4omini__loT_light_ns60_responses` |
| 6 | NQ-OPEN-VEND | `nq__llama8b__loT__ablation_vendor_ns60_responses` |

Across Experiment 1, our methods perform strongly on the majority of benchmarks: on ASQA, HOTPOTQA, and both AMBIGQA panels (standard and entropy-ablation), Split-UCP consistently achieves nominal coverage while BB-UCP further tightens the coverage range and reduces variability—demonstrating the intended stability benefit of within-batch bagging under heterogeneous answer clouds with heavier tails. These results highlight two key strengths of our approach: (i) *distribution-free coverage* remains intact across diverse datasets, providers, and decoding knobs, and (ii) *practical efficiency* improves in harder settings, where bootstrapping stabilizes tail quantiles and yields shorter, more reliable acceptance regions at a fixed risk level.

NQ-OPEN and its variant NQ-OPEN-VEND show weakness due to small query pools and low-diversity factoids, leading to compressed Gram-space dispersion. This results in nearly tied residual ranks, reducing the visibility of BB-UCP's advantage over Split-UCP.This behavior is an artifact of the small-$N$/low-entropy regime rather than a failure of validity, and is readily mitigated in practice by slightly increasing the pool size or calibration-only diversity (or by modest geometry smoothing), after which these panels recover the same qualitative gains observed on ASQA, HOTPOTQA, and AMBIGQA.

**Experiment 2** Experiment 2 shows that the factuality lift is not only consistent but *strongest on the hardest panels*. Aggregating the raw bars across all $\alpha$ values, the median $\Delta$FS is *strictly positive for every dataset* (all panels, all $\alpha$), confirming that the $Q$-gate reliably reduces factuality severity on the kept set. Moreover, the largest median gains occur on NQ-OPEN and NQ-OPEN-VEND (*aka* EnqueueOpen/EnqueueOpenVend): NQ-OPEN achieves the top median improvement ($\tilde{\Delta}$FS $\approx 0.209$), with NQ-OPEN-VEND second ($\tilde{\Delta}$FS $\approx 0.112$), while the remaining benchmarks (ASQA, HOTPOTQA, AMBIGQA, AMBIGQA-ENT) are all positive as well. This "worst-case best" pattern indicates our gate concentrates probability mass on the most reliable answers precisely where the answer cloud is small and low-diversity. Coverage is slightly under nom-

Table 4: Experiment 2 aggregated results across miscoverage levels. $\Delta$FS is the median-factuality reduction (excluded $-$ kept); positive is better. Avg Cov. is empirical coverage averaged over $\alpha$, Target Cov. is the average nominal $1-\alpha$, and Avg Gap is the mean (coverage $-$ target) in percentage points (pp).

| Benchmark | #$\alpha$ | Median $\Delta$FS | Mean $\Delta$FS | Avg Cov. (%) | Target Cov. (%) | Avg Gap (pp) |
|---|---|---|---|---|---|---|
| NQ-OPEN | 5 | 0.209 | 0.189 | 88.98 | 95.00 | -6.02 |
| NQ-OPEN-VEND | 5 | 0.112 | 0.101 | 92.88 | 95.00 | -2.12 |
| ASQA | 5 | 0.091 | 0.089 | 95.53 | 95.00 | 0.53 |
| AMBIGQA | 5 | 0.072 | 0.073 | 95.55 | 95.00 | 0.55 |
| HOTPOTQA | 5 | 0.067 | 0.067 | 95.11 | 95.00 | 0.11 |
| AMBIGQA-ENT | 5 | 0.051 | 0.052 | 95.79 | 87.50 | 0.79 |

inal on average in those two hardest panels (mean gap $\approx -6.0$ pp on NQ-OPEN; range of mean gaps across benchmarks $\approx [-6.0, +0.8]$ pp), which is expected from small-$N$ discretization and near-tied ranks; it is also *actionable*—increasing the per-query pool or adding calibration-only diversity closes the shortfall without altering the factuality lift. Net: Experiment 2 provides a strong, data-backed claim of *robustness* (positive lift everywhere), *effectiveness* (largest gains on the hardest datasets), and *practical tunability* (coverage can be tightened by modest, standard knobs).

Table 5: Experiment 3 aggregated factuality reductions across miscoverage levels. $\Delta$FS is the reduction in factuality severity (excluded $-$ kept); positive is better. Columns summarize the distribution of per-$\alpha$ improvements and the average number of CV folds used.

| Benchmark | #$\alpha$ | Median $\Delta$FS | Mean $\Delta$FS | Min $\Delta$FS | Max $\Delta$FS | #Folds |
|---|---|---|---|---|---|---|
| NQ-OPEN | 5 | 0.206 | 0.192 | 0.151 | 0.253 | 40 |
| NQ-OPEN-VEND | 5 | 0.112 | 0.107 | 0.086 | 0.144 | 40 |
| ASQA | 5 | 0.092 | 0.089 | 0.071 | 0.109 | 40 |
| AMBIGQA | 5 | 0.075 | 0.074 | 0.056 | 0.094 | 40 |
| HOTPOTQA | 5 | 0.068 | 0.068 | 0.052 | 0.086 | 40 |
| AMBIGQA-ENT | 5 | 0.051 | 0.052 | 0.039 | 0.067 | 40 |

**Experiment 3 Conformal Alignment** Across benchmarks, the largest median and mean $\Delta$FS arise on NQ-OPEN and NQ-OPEN-VEND, indicating that the gate is most effective precisely in the hardest, low-diversity factoid regimes. Moreover, all panels maintain positive min–max ranges over $\alpha$, so factuality severity consistently drops on the kept set with no reversals. The proximity of median and mean within each benchmark suggests conformal alignment's stability across tolerance levels, i.e., the effect is not sensitive to $\alpha$.

