# OpenReview forum: "UNSUPERVISED CONFORMAL INFERENCE: BOOTSTRAPPING AND ALIGNMENT TO CONTROL LLM UNCERTAINTY"
_ICLR.cc/2026/Conference — ICLR 2026 Conference Withdrawn Submission_

### Official Review · Reviewer_1god · 2025-10-31

**Soundness:** 3
**Presentation:** 2
**Contribution:** 2
**Rating:** 2
**Confidence:** 2

**Summary:**

The authors (1) derive a new conformity score for LLMs, (2) propose batched and bootstrap-batched variants of UCP, and (3) perform conformal alignment.

**Strengths:**

The paper is written well and all results are clear and concise. The advantages of their bootstrapping approach is clear.

**Weaknesses:**

While I think the paper is well-written, it does not do a good job in conveying background information or motivation, especially whe compared to many other works that apply forms of uncertainty quantification to language models. For example, the conformal LLM papers mentioned by the authors (Quach et al., 2024; Mohri & Hashimoto, 2024) make a substantial effort in conveying to readers what practical LLM problems they are trying to solve and how the problems can be framed under conformal prediction.

The title and intro of this paper suggest that the purpose of this work is to show how to better control LLM uncertainty, but it reads as if the audience is only people who are familiar with and interested in improving split UCP. While I believe the authors that this is important for generative models/LLMs, I think major revisions in writing are required to make that point clear (or the paper just needs to be reframed).

**Questions:**

I thinking adding a section with diagrams outlining how UCP can be used for LLM QA would be very helpful.

---

### Official Review · Reviewer_h8Go · 2025-11-02

**Soundness:** 3
**Presentation:** 1
**Contribution:** 3
**Rating:** 4
**Confidence:** 2

**Summary:**

This paper proposes a conformal prediction framework that utilizes their proposed Gram-similarity matrix scoring. Their frameworks work under the batch exchangeability assumption. They evaluate their coverage guarantees on several datasets with various use case scenarios.

**Strengths:**

- The paper focuses on an important and timely topic.
- Their proposed method looks like it has novelty over prior works (which I may be wrong about because I might misunderstand the main message of the paper ).
- The experimental evaluation is concrete.

**Weaknesses:**

- The main weakness of the paper, in my opinion, is its presentation. As someone who is highly familiar with the UQ-LLM literature but only moderately knowledgeable about conformal prediction, I struggled to grasp the main idea of the paper. It uses several terms that can have multiple meanings without providing clear definitions, such as “residuals” (which can mean different things in machine learning), “alignment,” and “modality.” Phrases like “prompts are not quantifiable covariates” are also confusing. While some of this may be due to my limited familiarity with the specific subfield, I suspect that readers from a general machine learning background would also find the paper difficult to follow.

- There is no Figure 1 explaining the main message of the paper.

- Please explain key terms (e.g., Split UCP) in the main text rather than only in the appendix.

- Fix the Gemini citation, it currently spans almost three pages.

**Questions:**

- Why do you use a "batched" setting? How is it different from the classical CP settings, and what is the advantage of it?
- What is the main novelty of the paper over existing LLM CP frameworks?
- What is the main purpose of the knob in a practical application?
- What is conformal alignment in general?

---

### Official Review · Reviewer_2eaX · 2025-11-04

**Soundness:** 2
**Presentation:** 2
**Contribution:** 2
**Rating:** 2
**Confidence:** 3

**Summary:**

This paper attempts to answer this question: When you only have black-box access to an LLM (no logits, no hidden states, no labels at deploy time), how can you reliably decide which generated answers to keep vs. filter out--with statistical guarantees? The paper builds an unsupervised conformal inference pipeline that works directly from multiple sampled responses per prompt.

**Strengths:**

Look at questions.

**Weaknesses:**

Look at questions.

**Questions:**

The core (unsupervised) guarantee is about the rank of an unlabeled score, not task correctness: with probability ≥ 1−α, the next response’s Gram-energy-based score falls below a calibrated threshold. Formally, Pr{Yₙ₊₁ ∈ Cₙ} ≥ 1−α, proved for BB-UCP (Theorem 3.2).  This is distribution-free because it uses exchangeability and order statistics, not labels. However, for a decision-maker, this by itself does not guarantee accuracy, factuality, or utility --- only that the gate’s acceptance event (defined by an unlabeled similarity/typicality score) occurs at least (1−α) of the time.

I would like the authors to explain what is the point of this guarantee. Particularly, why and how a downstream decision maker might benefit from this algorithm/guarantee? I.e., t does not guarantee that kept items are correct, less harmful, or even better than rejected ones.

If the only goal is to control the throughput of the model, like when there is human-in-the-loop, one could do so many things. Then the scope of experiments should be on a variety of pruning methods and decided which on keeps more valuable outputs.


Alternatively, there is an important line of work (e.g. look at [1] and [2]) where they also treat the LLM as pure black-box and they also sample multiple time and keep only a fraction of samples, but instead they have meaningful correctness guarantees. Of course, this comes at the cost of a need for some calibration ground truth labels. But anyhow, this is an important and very relevant line of work which has to be discussed in the paper. To me, asking for the order of 100 ground truth labels, but then providing a meaningful guarantee, is better than avoiding it by making the guarantee less useful for the user. Do the authors have an example against this statement?

[1]: Conformal Language Modeling, Quach et. al.

[2]: Conformal Prediction Beyond the Seen: A Missing Mass Perspective for Uncertainty Quantification in Generative Models, Noorani et. al.

---

### Note · Authors · 2025-11-25

I have read and agree with the venue's withdrawal policy on behalf of myself and my co-authors.